# *Haplocauda*, a New Genus of Fireflies Endemic to the Amazon Rainforest (Coleoptera: Lampyridae) [note 1]

**DOI:** 10.3390/insects13010058

**Published:** 2022-01-05

**Authors:** Luiz Felipe Lima da Silveira, William Lima, Cláudio Ruy Vasconcelos da Fonseca, Joseph McHugh

**Affiliations:** 1Biology Department, Western Carolina University, 460 Memorial Drive, Cullowhee, NC 28723, USA; 2Laboratório de Sistemática e Ecologia de Coleoptera (LASEC), Coordenação de Biodiversidade (CBIO), Instituto Nacional de Pesquisas da Amazônia (INPA), Manaus 69067-375, AM, Brazil; willgolima@gmail.com (W.L.); claudioruy.fonseca@gmail.com (C.R.V.d.F.); 3Department of Entomology, University of Georgia, 455 Biological Sciences Building, 120 Cedar Street, Athens, GA 30602, USA; mchugh.jv@gmail.com

**Keywords:** Lampyrinae, Photinini, Lucidotina, *Scissicauda*, copulation clamps

## Abstract

**Simple Summary:**

The Amazon Rainforest is still a frontier in the study of insect biodiversity, housing several species yet to be described and studied. Yet, its continuous deforestation urges scientists to study insect diversity there before it is too late. Here, we identified and described new species and a new genus of fireflies from the Amazon Rainforest. These new species are unique among other fireflies due to their unique abdominal morphology, supposed here to involve a clamping mechanism used during mating.

**Abstract:**

Most firefly genera have poorly defined taxonomic boundaries, especially in the Neotropics, where they are more diverse and more difficult to identify. Recent advances that shed light on the diversity of fireflies in South America have focused mainly on Atlantic Rainforest taxa, whereas lampyrids in other biomes remained largely unstudied. We found three new firefly species endemic to the Amazon basin that share unique traits of the male abdomen where sternum VIII and the pygidium are modified and likely work as a copulation clamp. Here we test and confirm the hypothesis that these three species form a monophyletic lineage and propose *Haplocauda*
**gen. nov.** to accommodate the three new species. Both maximum parsimony and probabilistic (Bayesian and maximum likelihood) phylogenetic analyses confirmed *Haplocauda*
**gen. nov.** monophyly, and consistently recovered it as the sister group to *Scissicauda*, fireflies endemic to the Atlantic Rainforest that also feature a copulation clamp on abdominal segment VIII, although with a different configuration. We provide illustrations, diagnostic descriptions, and keys to species based on males and females. The three new species were sampled from different regions, and are likely allopatric, a common pattern among Amazonian taxa.

## 1. Introduction

Beetles are the most diversified lineage of organisms, with over 400,000 species described, and over 1,000,000 are estimated [1]. Elateroidea is a beetle lineage that includes hard-bodied forms (e.g., click beetles), but also several lineages that independently evolved soft-bodiedness [2], a complex suite of traits that entails leathery elytra, exposed membranes between sclerites, and six to eight visible abdominal sternites. Among soft-bodied elateroids, fireflies are arguably the most popular.

For such charismatic and fascinating insects, fireflies can be surprisingly hard to identify. The study of lampyrid diversity has been hampered by a lack of taxonomic revisionary work and diagnostic resources to facilitate species identification, particularly in the Neotropics, where the family is most diverse. Despite recent advances in the classification at higher levels [3], the genus-level classification of most fireflies remains unchanged from pre-Hennigian taxonomy for most subfamilies. A notable exception to this static generic classification is seen in the Australopacific luciolines, which have a much more mature taxonomic status due to decades of revisionary studies and, more recently, phylogenetic analyses [4]. Otherwise, revisionary work for most genera across firefly subfamilies is needed, and the monophyly of most genera remains largely untested. These issues in combination contribute to the confusion surrounding the membership and diagnoses of most genera.

Recently, there has been a surge of new efforts to improve lampyrid taxonomy in South America through taxonomic reviews and the generation of identification keys (e.g., [5,6,7,8,9]), and through the incorporation of phylogenetic analyses in monographic treatments [9,10,11]. Most of this work, however, focused on the Southeastern Atlantic Rainforest taxa. In fact, firefly diversity across most South American biomes remains largely underestimated, since targeted sampling, e.g., [9], and museum-based research often reveal many new species. Given that the Amazon is the cradle of several important lineages of organisms (e.g., [12]), including taxa from that biome is of utmost relevance to the study of biodiversity and evolution. What is more, the fast pace of its deforestation makes the study of the Amazon basin urgent [13]. During one such study of museum material, we found three unusual, undescribed species that represent a potential new genus. These three species bear modifications of the terminalia that are similar to ones seen in *Scissicauda* McDermott, 1964, which are thought to work as copulation clamps [6].

Here, we formally describe and illustrate these three distinctive new species, provide keys to identify them based on both sexes, and examine their phylogenetic placement in order to determine whether they represent a monophyletic group worthy of generic-level rank. The three new species are presumed to be endemic to the Amazon Rainforest, where they were sampled across different regions, and are likely to be allopatric.

## 2. Materials and Methods

### 2.1. Morphology, Terminology, and Map

Specimens of these distinctive new species were located at the following institutions: the Brazilian National Institute of Amazonian Research (INPA) and University of Georgia Collection of Arthropods (UGCA); Museu de Zoologia da Universidade de São Paulo, São Paulo, Brazil (MZUSP); Colecão de Entomologia Pe. Jesus Santiago Moure, Universidade Federal do Paraná, Curitiba, Brazil (DZUP); Coleção de Entomologia Prof. José Alfredo Pinheiro Dutra, Universidade Federal do Rio de Janeiro, Rio de Janeiro, Brazil, (DZRJ); and the Coleção de Entomologia do Instituto Oswaldo Cruz (CEIOC). The abdomen was removed from 2 male and 1–2 female specimens per species and soaked in 10% KOH for 24–36 h. This clearing procedure was also applied to two entire specimens representing one of the three new species. The morphology was examined under a Leica M205 C stereomicroscope, and photographs were made with the Leica Application Suite X Auto-montage Software. We follow the classification of [3] and the anatomical terminology of [6]. Distribution maps of the species were made using QGIS 3.10.14 [14]. We recorded label data for all type specimens using the following conventions: double quotes (“ ”) for label data quoted verbatim, double forward slashes (//) to separate labels; double comma (,,) for line breaks, and brackets [ ] to enclose our comments or notes. All labels are typed unless otherwise noted.

### 2.2. Phylogenetic Analyses

We ran phylogenetic analyses with two goals: (i) to test the hypothesis that the three new species bearing the distinctive abdominal morphology constituted a monophyletic group, and, if so, (ii) to explore the placement of that group within Lampyridae. Of particular interest were the phylogenetic relationships of the new species with Lampyrinae Rafinesque, 1815, a subfamily that includes several taxa sharing similar anatomical features with the new species (see below).

Lampyrid classification remains unstable despite recent advances (e.g., [3]), and relationships among genera, as well as their monophyly, remain largely unexplored outside Luciolinae [4] (but see [10,11,15]). In order to determine the placement of these new species in Lampyridae, we included in our taxon sampling representatives of the four subfamilies that are known to occur in South America, namely, Psilocladinae (*Psilocladus miltoderus* Blanchard, 1846), Amydetinae (*Amydetes fastigiata* Illiger, 1807), and Lampyrinae, in addition to the taxa *incertae sedis* (*Araucariocladus hiems* Silveira & Mermudes, 2017, and *Vesta thoracica* Olivier, 1790). We included a denser sampling of Lampyrinae because the new species shared many similarities with the lampyrine *Costalampys* Silveira, Roza, Vaz & Mermudes, 2021, and *Scissicauda* McDermott, 1964, which was transferred to Lampyrinae in [3]. Data for the material examined of the new species is given below (see Section 3), and additional material used in the phylogenetic analyses is provided in Appendix A.

Using MESQUITE [16], we built a matrix with 20 taxa scored for 76 characters following the guidelines of [17]. The data include 55 characters reanalyzed from [15] combined with 21 new characters (see Section 3; Appendix A). Measurement-based characters are taken at the longest or the widest point of the respective structure. Key character states are labeled in figures, abbreviated as C:S, where C and S indicates character and state number, respectively, followed by the numbers of interest.

In order to compare the outcomes of multiple phylogenetic approaches, we performed maximum parsimony (MP) analyses in TNT [18], Bayesian inference (BI) in MrBayes 3.2.7a [19,20] (available at The CIPRES Science Gateway V. 3.3 [phylo.org]–[21], and maximum likelihood inference in IQTREE 1.6.12 [22].

We ran the MP analyses using New Technology heuristic searches with tree bisection and reconnection, with equal and implied weights (EW and IW, respectively), scaling the *k* parameter from 0.5 to 50 to investigate how homoplasy impacts tree topology [23,24]. Node support was assessed using the Bremer index and standard bootstrapping (for MP–EW), and symmetric resampling (for MP–IW) with 1000 replicates. Character evolution was optimized using WINCLADA [25].

The BI analysis implemented the MKV model, modified from MK to account for ascertainment bias due to the prior exclusion of invariant characters in our matrix [26], in addition to among-character rate variation with 8 rate categories for tractability [27]. We ran 50,000,000 generations, saving trees every 2000 generations and discarding the first 25% as burn-in. We checked for convergence using Tracer v1.6 [28]. We compared the fit and the topologies of the BIMKV with gamma and lognormal rate distribution, since each was found to provide dataset-specific improvements to evolutionary models by [29]. For the ML, we obtained estimates of branch support with ultrafast bootstrapping with 1000 replicates in IQTREE. To compare the three main topologies obtained (i.e., MLG (shared with MPEQ), BIG, and BILN), we performed pairwise comparisons in IQTREE, and applied the SH [30] and AU [31] statistical tests.

Trees were read in FigTree version 1.4.4 (obtained at https://github.com/rambaut/figtree/releases; accessed on 2 January 2021), and the different measures of node support obtained (Bremer index, bootstrap, ultrafast bootstrap, and posterior probabilities) were placed on the preferred topology (see below) using Adobe Photoshop 2021.

## 3. Results

### 3.1. Characters

Our matrix included characters from male specimens, spanning the three tagmas: head (12), thorax (15), and abdomen (49). Of note, aedeagal traits yielded 20 characters. The full character list is given below:Antenna, antennomeres III-IX, shape: (0) serrate (Figure 4G), (1) cylindrical.Antennal serration, shape: (0) acute (Figure 4G), (1) rounded.Antenna, antennomeres III-IX, single lamella: (0) absent, (1) present.Antenna, antennomeres III-IX, double lamellae: (0) absent, (1) present.Antenna, single flabellar insertion, position: (0) basal, (1) apical, (2) progressively apical.Antenna, upright bristles: (0) absent (Figure 4G), (1) present.Labrum, connection to frons: (0) connected by membrane (Figure 4C), (1) fused.Frons, shape: (0) concave, (1) flat (Figure 4A,C,D).Mandibles, orientation in frontal view: (0) largely overlapping (Figure 4C), (1) crossed.Gula, length relative to submentum: (0) at least 1/5 shorter (Figure 7B), (1) as long as, (2) at least 1/5 longer.Labium, submentum, shape: (0) subcordiform (Figure 4B), (1) U-shaped, (2) triangular, (3) tongue-shaped.Labium, palp, palpomere III, shape: (0) cylindrical and apically rounded; (1) with sides slightly arcuate, apex strongly emarginate; (2) with sides divergent, apex straight to slightly emarginate (Figure 7C,D).Pronotum, disc, convexity relative to explanate margins in lateral view: (0) flat (Figure 5C), (1) convex.Pronotum, lateral expansions, puncture depth: (0) deep, (1) shallow (Figure 5A).Pronotum, anterior margin, shape: (0) acuminate anteriorly, (1) evenly arcuate (Figure 5A).Pronotum, ratio between lateral expansion and disc widths: (0) less than 1/3, (1) almost 1/2 (Figure 5A).Pronotum, posterolateral angle, minute outwards projection: (0) present, (1) absent (Figure 5A,B).Pronotum, disc, sagittal depression: (0) absent (Figure 5A,D,E), (1) present.Pronotum, posterior expansion, posterior margin, shape in dorsal view: (0) strongly sinuate, (1) almost straight (Figure 5A,B).Hypomeron depth relative to width of the lateral expansion of pronotum: (0) about as long, (1) at least 1/3 shorter (Figure 5E).Prosternum, anterior margin, shape: (0) strongly sinuate medially, (1) almost straight (Figure 5B).Mesoscutellum, posterior margin, shape: (0) rounded (Figure 5F), (1) truncate.Elytron, outer margin, shape: (0) straight, (1) rounded (Figure 5L).Elytron, distinct black outline: (0) absent, (1) present (Figures 3 and 5L).Wing, mp-cu crossvein, position relative to the MP3+4 split: (0) more basal (Figure 5M), (1) more apical.Wing, AA3 vein, shape: (0) short (almost as long as wide) and almost perpendicular to AA4 (Figure 5M), (1) elongated and with an acute angle to AA4.Wing, r3 crossvein: (0) absent, (1) present (Figure 5M).Terga II-VII, posterior angles, shape: (0) acute (Figure 6A,B), (1) right-angled.Sternum VI, lantern: (0) absent (Figures 3 and 6C), (1) present.Sternum VI, lantern, length relative to sternum: (0) less than 1/3, (1) more than 1/3.Sternum VIII, core length (i.e., excluding the length of median projections) relative to VII: (0) as long as, (1) at least 1/5 longer (Figures 3 and 6C).Sternum VIII, posterior margin, tiny mucronate posterior projection: (0) absent (Figure 6C,D), (1) present.Sternum VIII, posterior margin, median third, long and wide posterior projection: (0) absent, (1) present (Figure 6C–E).Sternum VIII, posterior margin, apex of wide posterior projection, shape: (0) rounded (Figure 11J), (1) acute (Figure 6D,E).Pygidium, posterior margin, medial indentation: (0) absent (Figure 6F), (1) present.Pygidium, length/width ratio: (0) at least 1/5 wider than long, (1) as wide as long, (2) at least 1/5 longer than wide (Figure 6F).Pygidium, posterior margin, parasagittal indentations/sinuosities: (0) absent, (1) present (Figure 6F).Pygidium, lateral margins, shape: (0) subparallel, (1) rounded (Figure 6F), (2) divergent posteriorly, (3) convergent posteriorly.Pygidium, posterior angle length relative to central third: (0) shorter, (1) as long as, (2) longer (Figure 6F).Pygidium, postero-lateral angles, degree of development: (0) well-developed (Figure 6F), (1) barely conspicuous.Pygidium, anterior angles, blunt anterior projections: (0) absent, (1) present (Figure 6F).Pygidium, anterior margin, shape: (0) slightly emarginate, (1) strongly emarginate (Figure 6F).Syntergite, anterior margin, indentation: (0) absent, (1) present (Figure 6G).Syntergite, anterior margin, degree of indentation: (0) mild, (1) strongly indented, (2) completely separating the syntergite in two elongated sclerotized plates (Figure 6G).Syntergite, sagittal membranous suture: (0) absent (Figure 6G), (1) present.Syntergite, posterior-apical connection with sternum IX: (0) separated (Figure 6G,H), (1) fused.Syntergite, transverse keel, presence: (0) absent, (1) present (Figure 6G).Syntergite, transverse keel, position: (0) close to the posterior margin, (1) following contour of anterior margin (Figure 6G).Syntergite, length relative to sternum IX: (0) 1/4, (1) 1/2, (2) 2/3 (Figure 6G,H).Sternum IX, lateral rods, shape: (0) subparallel-sided, (1) convergent anteriorly (Figure 6H).Sternum IX, length relative to aedeagus (including phallobase): (0) at least slightly shorter, (1) slightly longer, (2) 1/3 longer (Figure 6H,I).Sternum IX, anterior/proximal distance between lateral rods: (0) widely separated, (1) gradually close-set, (2) fused (Figure 6H), (3) abruptly close-set.Sternum IX, lateral rods, apical margins: (0) acuminate, as thick as core rod, (1) truncate and thickened anteriorly (Figure 6H).Sternum IX, posterior margin, median projection: (0) absent, (1) present (Figure 6D).Sternum IX, position relative to VIII: (0) completely covered (Figures 3 and 6C), (1) partially exposed.Sternum IX, posterior margin, shape: (0) rounded (Figure 6H), (1) acute.Phallobase, shape: bilateral symmetry: (0) asymmetrical, (1) symmetrical (Figure 6I).Phallobase, length relative to dorsal plate of phallus: (0) at least 1/4 shorter (Figure 6I), (1) as long as.Phallus, dorsal plate, connection to parameres: (0) separated, (1) medially fused (Figure 6I).Phallus, dorsal plate, median fusion to parameres, extent: (0) up to 1/4 of phallus length (Figure 6I), (1) nearly 1/2 of phallus length.Phallus, dorsal plate, apical half, lateral margins, shape: (0) evenly acuminate, (1) abruptly acuminate (Figure 6I), (2) sinuate, (3) subparallel-sided.Phallus, dorsal plate, subapical transverse groove: (0) absent, (1) present (Figures 6J and 9P).Phallus, dorsal plate, length relative to parameres: (0) at least 1/5 longer, (1) at least 1/5 shorter (Figure 6I–L).Phallus, dorsal plate, apical indentation: (0) absent, (1) present (Figure 9O).Phallus, dorsal plate, overall shape in lateral view: (0) straight, (1) bent dorsally, (2) bent ventrally (Figure 6I–L), (3) sinuate.Phallus, dorsal plate, basal abrupt constriction: (0) absent (Figure 6I–L), (1) present.Phallus, dorsal plate, subapical outer keel: (0) absent (Figure 6I–L), (1) present.Phallus, dorsal plate, subapical outer teeth: (0) absent (Figure 6I–L), (1) present.Phallus, dorsal plate, apical arms (of indented phallus), shape: (0) widely distanced and slightly convergent, (1) close-set and strongly convergent (Figure 6I–L).Phallus, ventral plate: (0) absent, (1) present (Figure 9R).Phallus, ventral plate, length relative to dorsal plate: (0) half as long, (1) as long as, to slightly longer, (2) 1/3 shorter (Figures 9R and 11Q).Paramere, subapical tooth: (0) absent (Figure 6I–L), (1) present.Paramere, apex, shape: (0) straight, (1) curved ventrally, (2) curved inwards (Figure 6I–L).Paramere, apex, subapical abrupt constriction: (0) absent, (1) present (Figure 6I–L).Paramere (lateral view), ventral projection: (0) absent (Figure 6I–L), (1) present.Phallus, dorsal plate, subapical transverse groove, depth: (0) shallow (Figure 6I), (1) deep.

### 3.2. Phylogenetic Analyses

All analyses (MPEW, MPIE, BILN, BIG, MLG) recovered a clade comprising the three new species, and that clade was consistently positioned as the sister group to *Scissicauda* (Figure 1). As such, the name *Haplocauda*
**gen. nov.** (described below) is introduced here to accommodate the three new species (described below as *H. albertinoi*
**sp. nov.**, *H. yasuni*
**sp. nov.**, and *H. mendesi*
**sp. nov.**). The following uncontroverted synapomorphies support the monophyly *Haplocauda*: posterior margin of sternum VIII with a long and wide posterior projection (33:1), anterior angles of pygidium with blunt anterior projections (41:1), lateral rods of sternum IX thickened and truncate anteriorly (53:1), posterior margin of sternum IX with a median projection (54:1), dorsal plate of phallus with close-set apical arms (69:1), and paramere apices curved inwards (73:2). *Haplocauda* is further supported by the following homoplasies: gula shorter than submentum (10:0), syntergite a third shorter than sternum IX (49:2), phallobase bilaterally symmetrical (57:1) and 1/4 shorter than dorsal plate of phallus (58:0).

The following uncontroverted synapomorphies support the clade *Scissicauda* + *Haplocauda*: pygidium with posterolateral angles extending beyond central third of posterior margin (39:2), with anterior margin strongly emarginate (42:1); sternum IX with lateral rods anteriorly fused (52:2), completely covered under VIII (55:0); and dorsal plate with a subapical groove (62:1) and bent ventrally (65:2). The *Scissicauda* + *Haplocauda*
**gen. nov.** clade was also supported by the following homoplasies: pronotum disc flat (13:0); pronotal posterior angles rounded, without a median projection (17:1); elytron with distinct black outline (24:1); core sternum VIII longer than VII (31:1); and pygidium at least 1/5 longer than wide (36:2). *Scissicauda*, the sister-group of *Haplocauda*
**gen. nov.** was supported by five uncontroverted synapomorphies (posterior margin of pygidium medially indentate (34:1), lateral margins of pygidium divergent posteriorly (38:2), syntergite posteriorly fused to sternum IX [46:1], posterior margin of sternum IX acute (56:1), and dorsal plate of phallus fused to parameres throughout nearly 1/2 its length (60:1), in addition to one homoplasy: antenna with upright bristles (6:1).

Although the *Scissicauda + Haplocauda*
**gen. nov.** clade was consistently recovered with strong support in all analyses, there was no consensus about the relationships of neighboring genera. Alternate topologies of these taxa included the following: (i) a polytomy with *Lucidota*, *Dadophora Costalampys*, and *Scissicauda* + *Haplocauda* (MPIE and BILN); (ii) *Dadophora* (*Costalampys* (*Scissicauda* + *Haplocauda*
**gen. nov.**) (BIG); and (iii) *Dadophora* + *Costalampys* (*Scissicauda* + *Haplocauda*
**gen. nov.**) (MLG and MPEQ, Figure 1 and Figure 2), which is the favored topology here, given that it was recovered under two criteria. These slightly conflicting topologies (i–iv) were never strongly supported by any parameter measured, and the pairwise topology comparisons did not find statistically significant differences from one another (Appendix A). Below these poorly resolved genera, the phylogeny was stable with the relationships, as illustrated in Figure 1 and Figure 2. *Ethra* and then *Cladodes* received poor support for their positions as the next two most closely related taxa, respectively. At the next most basal node, the strongly supported clade *Dilychnia + Vesta* was recovered, and the strongly supported clade *Psilocladus + Araucariocladus* was recovered at the basal node of the tree.

Within *Haplocauda*
**gen. nov.**, the relationships between species were either weakly supported or unresolved: MPEW (Figure 2) and MPIW failed to resolve relationships within the genus, whereas BIG, BILN, and MLG (Figure 1) recovered *H. albertinoi*
**sp. nov.**
*+ H. yasuni*
**sp. nov.** as a weakly supported clade with *H. mendesi* as the sister group.


**Taxonomy**



**Lampyrinae Rafinesque, 1815**



**Photinini Olivier, 1907**



***Haplocauda* Silveira, Lima, and McHugh gen. nov.**



**urn:lsid:zoobank.org:act:081FEB79-3C72-42FA-BAFE-CF8072B0277F**


(Figure 3, Figure 4, Figure 5, Figure 6, Figure 7, Figure 8, Figure 9, Figure 10, Figure 11 and Figure 12)

**Type species:***Haplocauda albertinoi* sp. nov., by original designation.

**Diagnosis:** Antennae serrated and without branches (Figure 4G), pronotum with posterior angle rounded (i.e., lacking a median projection) (Figure 5A), hypomeron short in lateral view (width of lateral expansion at least 1.5x greater than hypomeron depth) (Figure 5C), abdominal sterna VI and VII without lanterns (Figure 3). Color pattern (Figure 3): body overall brown, except for light brown pronotal lateral expansions, and a longitudinal light brown lateral stripe on elytron (outlined in brown or light brown); antennomeres IX–XI sometimes creamy white; legs light brown, sometimes with brown tibia and tarsus; pygidium sometimes with translucent lateral spots; sternum VIII with translucent lateral spots. Male: anterior claw of pro- and mesolegs with basal teeth (Figure 5N–S); pygidium with anterior margin strongly emarginated, with thick, sinuate, and apically blunt anterior projections, posterior margin without a median indentation, posterior angles acute (Figure 6F); sternum VIII at least 2x longer than VII, with a very long and wide posterior projection (at least as long as core sternum VIII) (Figure 6D,E); sternum IX with lateral rods basally fused and posteriorly thickened, posterior margin of sternum IX with an acute projection (Figure 6G,H); aedeagus with phallobase bilaterally symmetrical, with sides strongly convergent basally and at basal margin, dorsal plate of phallus basally fused to parameres, apically indented and with arms apically curved and approximate, lacking ventral rods, ventral plate rudimentary, restricted to a sclerotized piece outlining the ejaculatory duct (Figure 6I–L). Female: pygidium with posterior margin truncated or rounded (Figure 8D); sternum VIII as long as wide and spiculum ventrale long and slender, as long as 3/4 sternum, with posterior margin emarginated or deeply emarginated (Figure 8E,F); internal genitalia with a large and somewhat rounded spermatophore-digesting gland and a lump-like spermatheca, bursa copulatrix with paired elongated and weakly sclerotized plates, with a very long accessory gland (Figure 8J). Ovipositor with valvifers free, twisted basally, 3× longer than coxite; coxites convergent posteriorly, divided in distinct proximal and distal plates, both well-developed but weakly sclerotized; styli minute, sclerotized; proctiger plate elongate, weakly sclerotized (Figure 8G–I).

**Etymology.** The name *Haplocauda* is composed of *haplo*, a Greek word for “simple,” and *cauda*, a Latin word for “tail.” The name is a word play with *Scissicauda*, a closely related genus, which has a relatively more complex abdominal morphology. Gender feminine.

Description

**Head** entirely covered by pronotum when retracted (Figure 3). Head capsule about 1.5× wider than long, lateral margins slightly convergent posteriorly, posterior margin with two posterior parasagittal indentations (Figure 4A); slightly longer than tall and slightly swollen dorsally in lateral view by frons, vertex somewhat convex (Figure 4D). Antennal sockets reniform, slightly wider than distance between sockets, antennifer process conspicuous (Figure 4C). Antenna 11-segmented, scape constricted proximally, pedicel almost as long as wide and constricted basally, antennomeres III–X variably serrated and without lamellae, without upright bristles, antennomeres IX– or X–XI brown or creamy white, apical antennomere about as long as the subapical (Figure 4G). Frontoclypeus very wide, slightly curved (Figure 4C). Labrum connected to frontoclypeus by a membranous suture (Figure 4C); nearly 4× as wide as long, anterior margin evanescent (Figure 4A,C). Mandibles largely overlapping, long and slender, evenly arcuate, apex acute, without internal tooth, external margin sparsely setose in basal 1/2 (Figure 4A,C). Maxilla weakly sclerotized (Figure 4); stipe about 2× longer than wide, posterior margins truncated, palpi 4-segmented, palpomere III subtriangular, IV lanceolate, with internal margin covered with minute, dense bristles, almost 3× longer than III (Figure 4C). Labium weakly sclerotized (Figure 4); mentum completely divided sagittally, submentum sclerotized and bearing bristles, subcordiform, elongated, palpi 3-segmented, palpomere III securiform (Figure 4B). Gular sutures almost indistinct, gular bar transverse, nearly 3× as wider than long (Figure 4B). Occiput piriform, 1/3 narrower than head capsule (Figure 4F).

**Thorax** with pronotum semilunar, with posterior angle rounded, disc subquadrate in dorsal view, almost flat in lateral view, regularly punctured, punctures small and pubescent, with a line of distinct deep marginal punctures (Figure 5A); pronotal expansions well-developed, anterior expansion maximal length almost half as long as disc, posterior expansions straight (Figure 5A); slightly wider than distance between elytral humeri (Figure 3). Hypomeron short in lateral view (width of lateral expansion of pronotum at least 1.5x greater than hypomeron depth) (Figure 5E). Prosternum 4× as wide as its major length, slightly narrowed parasagittally (Figure 5B). Proendosternite elongated, about as long as distance between the apices of proendosternite arms (Figure 5B). Mesoscutellum with posterior margin rounded (Figure 5F). Elytron ellipsoid, almost 3.5× longer than wide, pubescent, with a row of conspicuous punctures surrounding sutural and lateral margins. Hind wing well developed, posterior margin slightly sinuate, 2× as long as wide, r3 slightly shorter than r4, radial cell 2× wider than long, almost reaching anterior margin, costal row of setae inconspicuous; CuA2 present, mp-cu crossvein present; RP + MP1 + 2 3/4 r4 length, almost reaching distal margin, J indistinct (Figure 5M). Allinotum slightly wider than long, lateral margins slightly convergent posteriorly, posterior margin straight, pubescent, prescutum extending slightly less than half metascutum length, rounded area of scutum weakly sclerotized, scutum-prescutal plates sclerotized, extending ridges almost to posterior margin, metascutellum sparsely pubescent (Figure 5H); anterior margin of alinotum trisinuate (Figure 5G). Mesosternum weakly sclerotized, acute medially, mesepimeron attached to metasternum by membrane, mesosternum/mesepisternum suture inconspicuous, mesepisternum/mesepimeron suture conspicuous (Figure 5I). Metasternum oblique and strongly depressed anteriorly (where the mesocoxae fit), anterior medial keel prominent up to anterior 1/3, discrimen indistinct, lateral margins divergent posteriorly up to lateral-most part of metacoxa, then convergent posteriorly, posterior margin bisinuate (Figure 5I). Profemur about as long as protibia (Figure 5N), meso and metafemora slightly shorter than respective tibiae (Figure 5O,P). Tibial spur formula: protibia 1 (only posterior), mesotibia 2, metatibia 1 (only anterior) or 2 (Figure 5Q–S). Male with anterior claw of pro and mesothoracic legs with basal tooth (Figure 5N–S). Tarsomere I 2× longer than II, II 2× longer than III, III subequal in length to IV, IV bilobed, lobes reaching 2/3 V length (Figure 5Q–S). Mesendosternum with two parasagittal projections directed outwards, irregularly alate (with flaps of irregular width) (Figure 5K). Metendosternum spatulate, 2× longer than wide, median projection acute anteriad, with two lateral laminae (Figure 5K).

**Abdomen** with tergum I with anterior margin membranous, laterotergite membranous, nearly rounded, with sparse bristles (Figure 5J); spiracle obliquely oriented on the thorax (Figure 5H,J). Terga II–VII with posterolateral angles more produced and acute posteriorly, posterior margins more bisinuate (Figure 6A). Sterna II–VIII visible (Figure 6C). Spiracles dorsal, at about 1/2 sterna lengths (Figure 6A).

**Male.** Sternum VIII 2x longer than VII, with a long and wide posterior projection (Figure 6D,E), with “larval” lanterns elongated (Figure 3). Pygidium with anterior margin strongly emarginated, with thick, sinuate, and apically blunt anterior projections; posterior margin without a median indentation; posterolateral angles acute (Figure 6F). Syntergite medially divided, bearing bristles posteriorly, not anteriorly fused to sternum IX, 1/3 shorter than sternum IX, connate to sternum IX along its length (Figure 6G,H). Sternum IX with lateral rods basally fused and posteriorly thickened, posterior margin of sternum IX with an acute projection (Figure 6G,H). Aedeagus with phallobase bilaterally symmetrical, sides abruptly convergent basally; dorsal plate of phallus basally fused to parameres, apically indented and with arms apically curved and approximate; parameres sinuate, nearly 1/5 longer than dorsal plate of phallus, lacking ventral rods; ventral plate rudimentary, restricted to a sclerotized piece outlining the opening of the ejaculatory duct (Figure 6I–L).

**Female.** Overall similar to male (Figure 3 and Figure 7), except for the following traits. Metatibia with two spurs. Pygidium with posterior margin truncate (Figure 8D) or rounded (Figure 12I). Sternum VIII as long as wide, spiculum ventrale long and slender, 3/4 sternum length, with posterior margin moderately indented or deeply indented (Figure 8E and Figure 10B). Internal genitalia with a large and somewhat rounded spermatophore-digesting gland and a lump-like spermatheca, bursa copulatrix with paired elongate, weakly sclerotized plates (Figure 10D–H), or with eight irregularly shaped well-sclerotized plates (Figure 8J) with a very long accessory gland (Figure 8J). Ovipositor with valvifers free, twisted basally, 3× longer than coxite; coxites convergent posteriorly, divided into distinct proximal and distal plates, both well developed but weakly sclerotized; styli minute, sclerotized; proctiger plate elongate, weakly sclerotized (Figure 8G–I and Figure 10E,F).

**Distribution.** Amazon Rainforest, South America (Figure 13).

**Remarks.***Haplocauda***gen. nov.** is placed in Photinini (Lampyrinae) due to their well-developed arcuate mandibles (Figure 4A,C), pronotal anterior and lateral expansions (Figure 3), abdomen with dorsally placed spiracles (Figure 6A), and eight visible sterna (Figure 6C) (see [32]). Among the Photinini included in our matrix, *Haplocauda*
**gen. nov.** was found to be closely related to *Costalampys* and *Scissicauda* (see above). Males of these three genera can be readily separated by morphological traits. *Haplocauda*
**gen. nov.** can be distinguished from *Costalampys* by the pronotal posterior angle without an acute projection (Figure 5A), in addition to the elongated pygidium and sternum VIII (each nearly 1/3 longer than wide), with the sternum VIII having a long and wide posterior projection (Figure 6A–F). In *Costalampys*, the pronotal posterior angle has a minute acute projection, the pygidium is as long as wide, and sternum VIII is 2× wider than long and is mucronate or mildly sinuate (and lacking a long and wide posterior projection) [15]. *Haplocauda*
**gen. nov.** can be distinguished from *Scissicauda* by the pygidium with sides convergent posteriorly from midlength, and the sternum VIII with a long and wide posterior projection (Figure 6A–F). In *Scissicauda*, the pygidium has lateral sides divergent posteriorly, and the sternum VIII is sinuate and has no posterior projection [6]. *Haplocauda*
**gen. nov.** is superficially similar to some species currently placed in *Lucidota*, a poorly defined neotropical Photinini genus. However, *Haplocauda*
**gen. nov.** can be distinguished from *Lucidota* species by the elongated pygidium and sternum VIII (these are nearly 1/3 longer than wide), and by the presence of a long and wide posterior projection on the sternum VIII (Figure 6A–F).

Due to a lack of detailed descriptive and comparative studies on lampyrine females, genus-level identification based on females remains elusive. Among South American Lampyrinae, females of *Haplocauda*
**gen. nov.** are morphologically more similar to *Costalampys* and *Scissicauda*. *Haplocauda*
**gen. nov.** females can be distinguished from those of *Costalampys* by the pronotum with a flat disc and the lack of an acute projection on the posterior angle (Figure 3) (convex and with acute projection in *Costalampys*; Silveira et al., 2021). Finally, *Haplocauda*
**gen. nov.** females can be distinguished from those of *Scissicauda* by their pygidium with the posterior margin rounded (Figure 12I) or truncate (Figure 8D) indented in *Scissicauda* [6].

### 3.3. Key to Haplocauda Species Based on Males

1-Antennomeres IX– or X–XI brown (Figure 11G,H); pygidium with central third emarginate (Figure 11I); sternum VIII with posterior projection rounded and not reaching posterior margin of pygidium (Figure 11J,K)...….………… ***Haplocauda mendesi* sp. nov.**

1′-Antennomeres IX– or X–XI creamy white (Figure 9G,H); pygidium with central third truncated (Figure 9I) to barely rounded (Figure 6F); sternum VIII with posterior projection pointed and reaching (Figure 6D,E) or extending beyond the posterior margin of pygidium (Figure 9J,K) …..……...…….…………………………………………………..2.

2-Elytron longer than 7 mm (Figure 3A,B); metatibia with two spurs (Figure 5S); pygidium lateral thirds mostly light brown to translucent (Figure 6F); sternum VIII mostly light brown to translucent, with posterior projection slender and evenly curved dorsally (Figure 6D,E) ……..……...…….………………………….... ***Haplocauda albertinoi* sp. nov.**

2′-Elytron shorter than 5 mm (Figure 3D,E); metatibia with one spur; sternum VIII brown except for anterolateral translucent spots, with posterior projection stout and abruptly curved dorsally near apex (Figure 9J,K); pygidium brown except for anterolateral translucent spots (Figure 9I) ……..……...…….……………….. ***Haplocauda yasuni* sp. nov.**

### 3.4. Key to Haplocauda Species Based on Females

1-Antenna entirely brown (Figure 12G,H), posterior margin of pygidium rounded (Figure 12I) ……..………………….………………..….……… ***Haplocauda mendesi* sp. nov.**

1′-Antennomeres IX– or X–XI creamy white (Figure 7G,H), posterior margin of pygidium truncated (Figure 8D) ……..……...…….…...…….…...…….…………..…...…….. 2

2-Elytron longer than 7 mm (Figure 3C,D); pygidium with anterior margin medially indented, central third brown, lateral thirds translucent (Figure 8D); sternum VIII largely translucent (Figure 8E) ….……..……………...…………...... ***Haplocauda albertinoi* sp. nov.**

2′-Elytron shorter than 5 mm (Figure 3G,H); pygidium with anterior margin evenly rounded (Figure 10A); sternum VIII entirely brown (Figure 10A,B) …. ***Haplocauda yasuni* sp. nov.**


***Haplocauda albertinoi* Silveira, Lima, and McHugh sp. nov.**



**urn:lsid:zoobank.org:act:ADE5A8D9-A740-4E1D-8A70-D37DF494A313**


(Figure 3, Figure 4, Figure 5, Figure 6, Figure 7 and Figure 8)

**Diagnostic description.** Antennomeres IX or X–XI creamy white (Figure 3G,H). Pygidium brown with lateral thirds mostly light brown to translucent (Figure 6F). Sternum VIII mostly light brown to translucent. **Male.** Metatibia with two spurs (Figure 5S). Pygidium with anterior blunt projections divergent anteriorly and with inner margin straight, with central third distinctly rounded (Figure 6F). Sternum VIII mostly light brown to translucent; posterior projection basally as wide as 1/3 sternum width, evenly slender, reaching or slightly extending beyond the posterior margin of pygidium, homogeneously curved dorsally (Figure 6C–E). **Female.** Pygidium with anterior margin medially indented, posterior margin truncated (Figure 8D). Sternum VIII with spiculum ventrale 1/3 shorter than sternum, posterior margin moderately indented (Figure 8E,F). Internal genitalia with spermatophore-digesting gland nearly as large as the spermatheca, bursa copulatrix with 4 pairs of irregularly shaped (somewhat rounded) well-sclerotized plates (Figure 8J).

**Etymology.** This species was named in honor of Dr. José Albertino Rafael, a Brazilian professor whose research is focused on the diversity of Amazonian insects. Dr. Rafael encouraged us to study Amazonian fireflies. He also collected and kindly provided several of the specimens studied in this paper.

**Type material. HOLOTYPE** (Figure 3A,B) (♂, INPA, pinned), label data: “BRAZIL, Acre: Bujari, FES Antimary, 09°20′01″ S–68°19′17″ W, 21.x-04. xi, 2016, Malaise grande, E. F. Morato, & J. A. Rafael cols–Rede BIA.//HAPLOCAUDA ALBERTINOI HOLOTYPE”. **PARATYPES:** (12 ♀, INPA- COL 001661, pinned), label data: “BRAZIL, Acre: Bujari, FES Antimary, 09°20′01″ S–68°19′17″ W, 03. Viii-08.ix, 2016, Malaise grande, E. F. Morato, & J. A. Rafael cols–Rede BIA.//HAPLOCAUDA ALBERTINOI PARATYPES [red label, W. Lima’s hand].” (34 ♂, 5 ♀, INPA; 2♂, CEIOC; 2♂, DZRJ; 2♂, DZUP; 2♂, MZUSP), label data: “BRAZIL, Manaus, Reserva Ducke, Malaise,, v. 1996, M. G. V. Barbosa. Manaus, ZF2, Km 4,, 02°35′21″ S–60°06′55″ W, 17-31.viii. 2016,, Malaise (perto de pequeno igarapé), J. A. Rafael. [handwritten, unknown writer]//HAPLOCAUDA ALBERTINOI PARATYPES [red label, W. Lima’s hand].” (1 ♂, INPA-COL- 0001661, pinned), label data: “BRASIL, Manaus, Reserva Ducke–Platô Leste-Oeste,, 25.xii.2006-11.i.2007, Malaise suspensa–Sub-bosque,, G. Freitas & M. Feitosa cols. [handwritten, unknown writer]//HAPLOCAUDA ALBERTINOI PARATYPES [red label, W. Lima’s hand].” (1♂, 1♀, INPA-COL 001663), label data: “BRAZIL, Manaus, Rod. Am. 010, Km 26, Reserva Ducke, ix.2001, J. F. Vidal/Malaise- Mata. [handwritten, unknown writer]//HAPLOCAUDA ALBERTINOI PARATYPES [red label, W. Lima’s hand].” (1♀, INPA-COL 001662), label data: “BRASIL, Manaus, Reserva Ducke–Platô Leste-Oeste,, 21.vii.2007, Malaise,, G. Freitas & M. Feitosa cols. [handwritten, unknown writer]//HAPLOCAUDA ALBERTINOI PARATYPES [red label, W. Lima’s hand].”

**Remarks.***H. albertinoi***sp. nov.** is superficially similar to *H. yasuni*
**sp. nov.** in having two or three creamy white apical antennomeres (entirely brown in *H. mendesi*
**sp. nov.**), the male sternum VIII posteriorly acuminated and reaching the posterior margin of the pygidium (rounded and shorter, not reaching the posterior margin of the pygidium in *H. mendesi*
**sp. nov.**) (Figure 3A–H). *H. albertinoi*
**sp. nov.** males have two metatibial spurs (Figure 5S) (*H. yasuni*
**sp. nov.** lack the anterior metatibial spur) and have a slender posterior projection of the sternum VIII that is evenly curved dorsally (Figure 6D,E) (stout and abruptly curved dorsally near apex in *H. yasuni*
**sp. nov.**; Figure 9J,K). Moreover, *H. albertinoi*
**sp. nov.** females have a medially indented anterior margin of the pygidium (Figure 8A) (evenly rounded in *H. albertinoi*
**sp. nov.**; Figure 10A).


***Haplocauda yasuni* Silveira, Lima, Fonseca, and McHugh sp. nov.**



**urn:lsid:zoobank.org:act:15A9E3AA-B5F2-4C7F-B294-65463386CD22**


(Figure 11 and Figure 12)

**Diagnostic description**. Antennomeres IX or X–XI creamy white (Figure 9G,H). Pygidium brown except for anterolateral translucent spots with central third truncated (Figure 9I). Sternum VIII brown except for anterolateral translucent spots (Figure 9J–K). **Male**. Metatibia with one spur (posterior spur lacking). Pygidium with anterior blunt projections divergent anteriorly and with inner margin sinuate, posterior margin with central third almost straight (Figure 9I). Sternum VIII brown except for anterolateral translucent spots, with posterior projection stout (about 1/2 as wide as sternum at base) and abruptly curved dorsally near apex (Figure 9J,K). **Female**. Pygidium with anterior margin evenly rounded, posterior margin truncated (Figure 10A). Sternum VIII with spiculum ventrale 1/2 as long as sternum, posterior margin deeply indented (Figure 10B). Internal genitalia with spermatophore-digesting gland almost 3x larger than spermatheca, bursa copulatrix with paired elongated and weakly sclerotized plates. (Figure 10G,H).

**Etymology.** The specific epithet is based on the Yasuní River, one of the most important tributaries of the Yasuní National Park and Biosphere Reserve, where the type specimen was collected. Noun in apposition.

**Type material: HOLOTYPE:** (1♂, card-mounted, head and abdomen dissected and stored in microvial, UGCA), label data: “ECUADOR Napo, Misahualli nr. Tena, 3–8 Oct. 1999, Steven R. Keller col.//HAPLOCAUDA YASUNI HOLOTYPE [red label, in L. Silveira’s hand].” **PARATYPES:** (1♂, pinned, UGCA), label data: “ECUADOR Napo, Yasuni Res[earch]. Sta[tion]., 30 September–11 October 2002, C. Brammer 250m, 076°23.851′ W 0°40.566′ S.//HAPLOCAUDA YASUNI PARATYPE [red label, in L. Silveira’s hand].” (1♂, 2♀, pinned UGCA) label data: “ECUADOR Napo Yasuni Res[earch]. Sta[tion]. 19–30 October 1998, W. J. Hanson 250m//6°36′ W 0°38′ S//HAPLOCAUDA YASUNI PARATYPE [red label, in L. Silveira’s hand].” (1♂, pinned, UGCA), label data: “ECUADOR Napo, Misahualli nr. Tena, 26 August–2 September 2000, Steven and Paul Keller//HAPLOCAUDA YASUNI PARATYPE [red label, in L. Silveira’s hand].” (1♀, pinned, UGCA), label data: “ECUADOR Napo, Misahualli nr. Tena, 6–19 October 2001, C. Brammer Mal[aise]. Tr[ap]//HAPLOCAUDA YASUNI PARATYPE [red label, in L. Silveira’s hand].”

**Remarks.***H. yasuni***sp. nov.** is superficially similar to *H. albertinoi*
**sp. nov.** in having two or three creamy white apical antennomeres (entirely brown in *H. mendesi*
**sp. nov.**), and the male sternum VIII posteriorly acuminated and reaching the posterior margin of the pygidium (rounded and not reaching the posterior margin of the pygidium in *H. mendesi*
**sp. nov.**) (Figure 3A–H). *H. yasuni*
**sp. nov.** males lack the anterior spur of the metatibia (present in *H. albertinoi*
**sp. nov.**; Figure 5S), and have a stout posterior projection of sternum VIII that is abruptly curved dorsally near the apex (Figure 9J,K) (slender and evenly curved dorsally in *H. albertinoi*
**sp. nov.**; Figure 6D,E). In *H. yasuni*
**sp. nov.** females, the anterior margin of the pygidium is evenly rounded (Figure 10A) (medially indented in *H. albertinoi*
**sp. nov.;** Figure 8A).


***Haplocauda mendesi* sp. nov. Silveira, Lima, and McHugh**



**urn:lsid:zoobank.org:act:B904DF0F-92CF-4F1B-AF94-6DAC370C7871**


(Figure 9 and Figure 10)

**Diagnostic description**. Antennomeres IX or X–XI brown (Figure 11G,H). Pygidium brown except for anterolateral translucent spots (Figure 11I). Sternum VIII brown except for anterolateral translucent spots (Figure 11J,K). **Male**. Metatibia with one spur (posterior spur lacking). Pygidium with anterior blunt projections subparallel-sided and with inner margin straight, posterior margin with central third emarginated (Figure 11I). Sternum VIII with posterior projection rounded, evenly curved in lateral view, and not reaching the posterior margin of the pygidium (Figure 11J,K). **Female**. Pygidium with anterior margin evenly emarginated, posterior margin rounded (Figure 12I). Sternum VIII with spiculum ventrale 1/2 as long as sternum, posterior margin moderately indented (Figure 12J,K). Internal genitalia with spermatophore-digesting gland almost 3x larger than spermatheca, bursa copulatrix with paired elongated and weakly sclerotized plates (Figure 12L–O).

**Etymology.** This species was named in honor of the late Brazilian environmentalist Francisco “Chico” Mendes, murdered by a rancher in December 1988. As a rubber tapper and trade union leader, he championed the preservation of the Amazon Rainforest, as well as the human rights of Brazilian peasants and indigenous peoples.

**Type material: HOLOTYPE:** (♂, card-mounted, head and abdomen dissected and stored in microvial, UGCA), label data: “BRAZIL Rondonia, 62 km SE Ariquemes, 5–16 November 1996, W. J. Hanson//HAPLOCAUDA MENDESI HOLOTYPE [red label, in L. Silveira’s hand].” **PARATYPES:** (2 ♂, pinned, UGCA), label data: “BRAZIL Rondônia, 62km SE Ariquemes, 1–14 November 1997, B. Dozier//HAPLOCAUDA MENDESI PARATYPE [red label, in L. Silveira’s hand].” (1 ♂, 2 ♀, pinned, UGCA), label data: “BRAZIL Rondônia, 62 km SE Ariquemes, 5–16 November 1996, W. J. Hanson//HAPLOCAUDA MENDESI PARATYPE [red label, in L. Silveira’s hand].”

**Remarks.***H. mendesi***sp. nov.** can be distinguished from the remaining *Haplocauda* species by antennae that are entirely brown (apical antennomeres are creamy white in other spp.) and males with a short sternum VIII not reaching the posterior margin of the pygidium (longer, reaching the posterior margin of the pygidium in other spp.), with the posterior projection apically rounded (truncated in other spp.).

## 4. Discussion

### 4.1. On the Fireflies of the Amazon Rainforest

Recent beetle research across South American biomes continues to find new taxa, some of which are of crucial importance to understanding the evolution of elateroid beetles. For example, a whole new family of beetles—Jurasaidae, elateroid beetles that convergently evolved soft-bodied morphology, and larviform females—was recently described from the Atlantic Rainforest [33]. New jurasaid species continue to be described, both fresh and museum specimens alike [34,35]. The recent finding of Jurasaids illustrates how much is yet to be discovered in South America. Despite significant advances in the systematics of Atlantic Rainforest lampyrids, the Amazon Rainforest continues to be a frontier in firefly research.

Several firefly species are known to occur in the Amazon Rainforest, and some are thought to be endemic to that biome (e.g., *Amydetes luzecu*, *A. marajoara*, and *A. vagalume* [5]). Given that most neotropical firefly species are known only from their original taxonomic descriptions, which often have vague type-localities (e.g., “Brazil,” “Nova Granada”), and given that most neotropical genera lack comprehensive reviews based on samples from multiple regions, the number of Amazon lampyrid endemics is likely to be severely underestimated. *Haplocauda*
**gen. nov.** is, to our knowledge, the first genus that is thought to be entirely endemic to the Amazon Rainforest. Even though the genus could exist undetected in other Neotropical biomes, it is not known from the Atlantic Rainforest (where the sister taxon *Scissicauda* occurs) despite extensive sampling there [9] and references therein.

### 4.2. On Haplocauda and Scissicauda

Historically, lampyrid taxonomy above the species level was largely based on a few traits involved in mate finding and courtship (i.e., form of sensory organs and lanterns). However, our study points out that such traits are variable within genera (e.g., presence of antennal branches), plesiomorphic (e.g., antennal serration), or invariable (e.g., absence of lanterns) among closely related genera in our dataset (e.g., *Haplocauda* and *Scissicauda*). Our findings support the hypothesis that traits involved in mate finding and courtship can be highly variable among closely related species, as also suggested in recent phylogenetic studies based on molecular data [3,36]. Furthermore, we found that some traits of the terminalia and genitalia support nodes above the species level (Figure 2), highlighting their value in the higher-level classification of lampyrid fireflies.

*Haplocauda* and *Scissicauda* were recovered as sister taxa using both parsimony and probabilistic phylogenetic methods (Figure 1 and Figure 2). The numerous synapomorphies supporting the monophyly of the *Haplocauda*, *Scissicauda* clade, and *Haplocauda* + *Scissicauda*, including newly delimited traits (see Section 3), highlight the value of detailed morphological data in firefly taxonomy.

### 4.3. On Firefly Copulation Clamps

Several firefly species have morphological specializations that are known or suspected to be used by males to clasp females, thereby functioning as copulation clamps that facilitate sperm transfer. These include toothed claws (e.g., *Micronaspis* [11]), pointed elytral apices (e.g., *Pteroptyx* [37]), aedeagus (phallic and parameral) teeth (e.g., *Ybytyramoan* [38]), and modifications of sclerites on some of the distal abdominal segments, including segment VII (e.g., *Pygoluciola* [39]), VIII (e.g., *Scissicauda* [6]), and both VIII and IX [8]. The scattered distribution of these traits across distantly related lampyrid taxa suggests that they were acquired independently. Sometimes multiple clasping traits occur together in one species; for example, *Scissicauda* and *Haplocauda* males have toothed claws and modifications of the sternum VIII and pygidium [6].

In *Luciuranus*, at least some species have species-specific male specializations that are matched by modifications of female abdominal segments VI–VIII; together these are thought to work as a lock-and-key system [7,40]. *Haplocauda* males have a specialized sternite VIII and pygidium (see above), which suggests that these are used to stabilize copulation. In this case, however, complementary anatomical specializations are not seen in conspecific females. Future field studies are sorely needed to clarify the functioning of firefly copulation clamps.

## 5. Conclusions

Here, we described three new species in a new genus of fireflies endemic to the Amazon basin with unique copulation clamps. Our finding underscores the need for more targeted studies across South American biomes and the value of biological collections in Biodiversity research.

## Figures and Tables

**Figure 1 insects-13-00058-f001:**
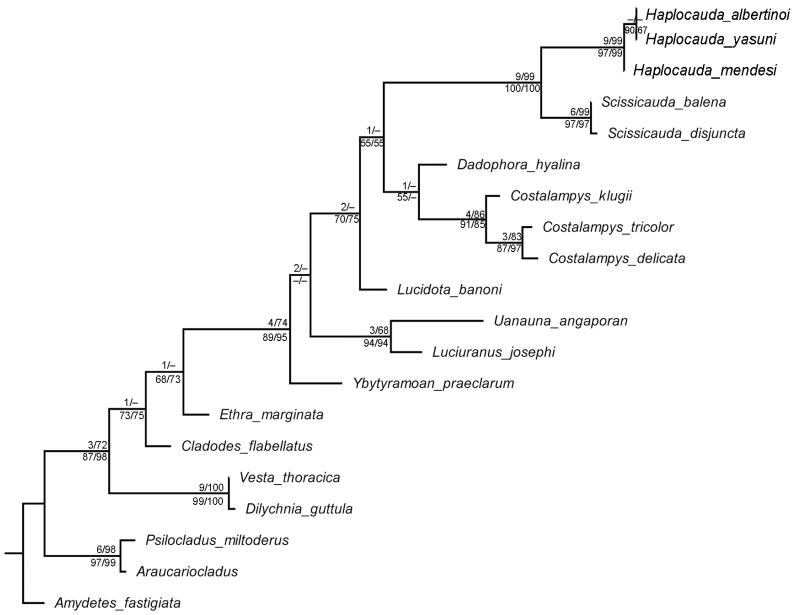
Preferred topology obtained with maximum likelihood (with MKV model) and concordant with maximum parsimony (with equal weights). Maximum parsimony Bremer index (left), bootstrap support values are given above nodes (right). Maximum likelihood UFBoot support (left) and Bayesian posterior probabilities (right) are given below nodes. Values below 50 are represented by a dash.

**Figure 2 insects-13-00058-f002:**
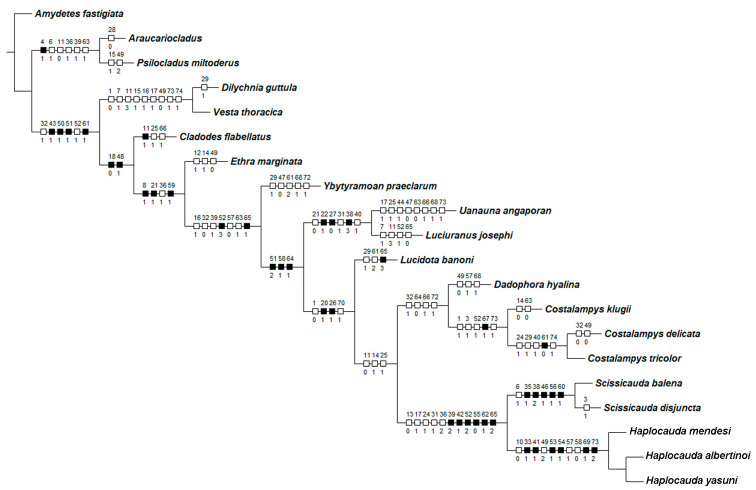
Uncontroverted pomorphies (black squares) and homoplasies (white squares) mapped on the preferred topology.

**Figure 3 insects-13-00058-f003:**
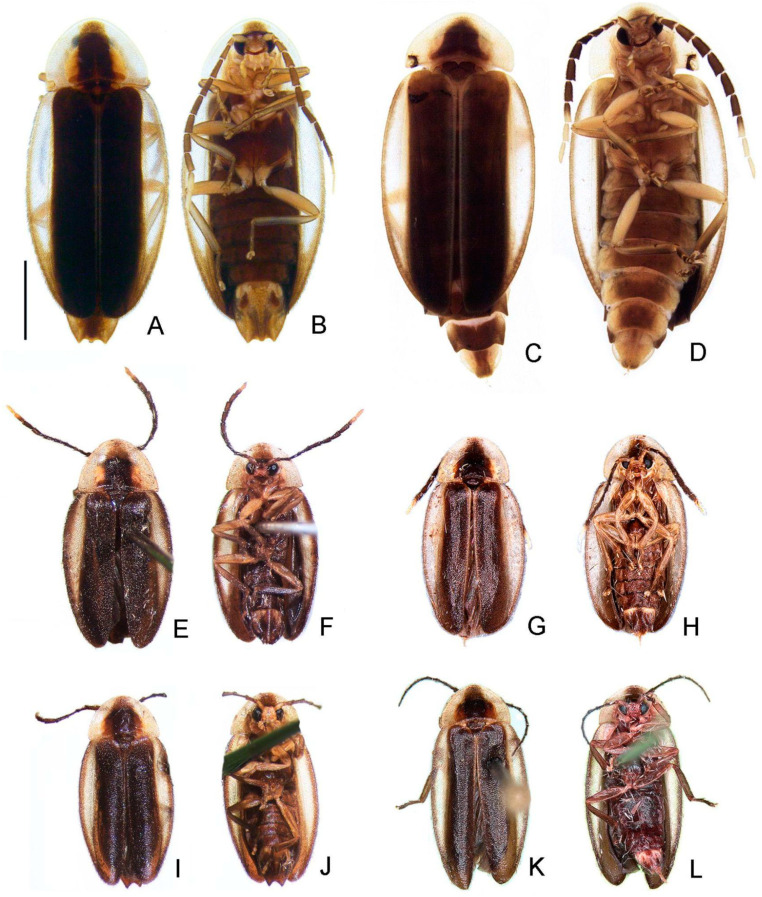
Habiti of *Haplocauda* spp. *H. albertinoi*
**sp. nov.**, male: (**A**) dorsal, (**B**) ventral; female: (**C**) dorsal, (**D**) ventral. *H. yasuni*
**sp. nov.**, male: (**E**) dorsal, (**F**) ventral; female: (**G**) dorsal, (**H**) ventral. *H. mendesi*
**sp. nov.**, male: (**I**) dorsal, (**J**) ventral; female: (**K**) dorsal, (**L**) ventral. Scale bar: A–L = 2.5 mm.

**Figure 4 insects-13-00058-f004:**
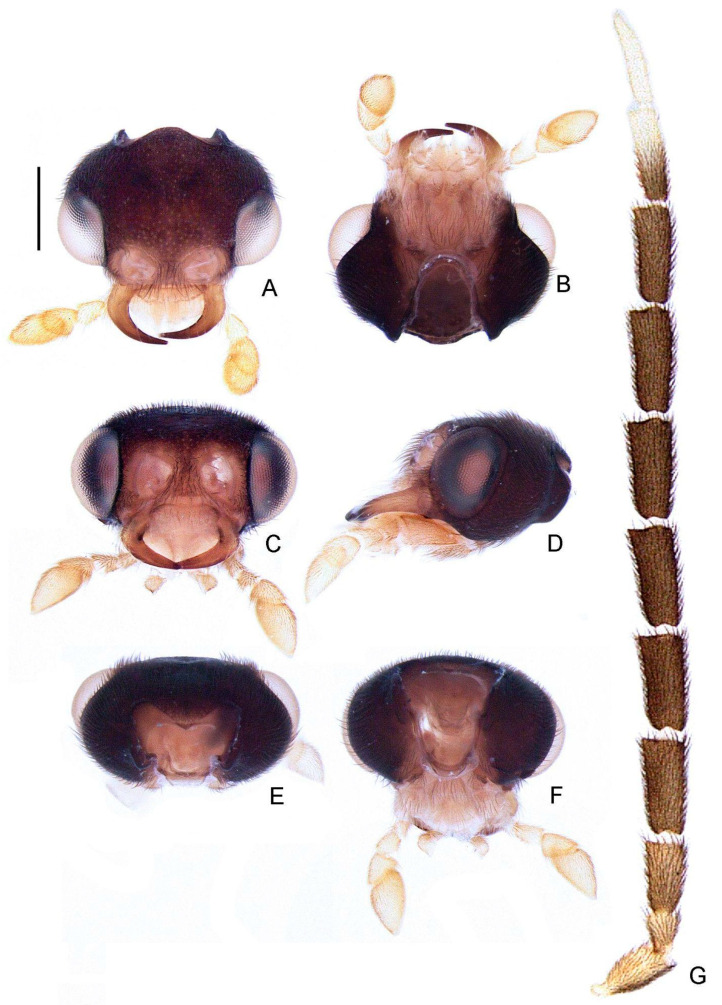
*Haplocauda albertinoi***sp. nov.**, male head. Head capsule (**A**–**F**): (**A**) dorsal, (**B**) ventral, (**C**) frontal, (**D**) lateral, (**E**) posterior, (**F**) occipital. (**G**) Left antenna, dorsal. Scale bar: A–G = 500 µm.

**Figure 5 insects-13-00058-f005:**
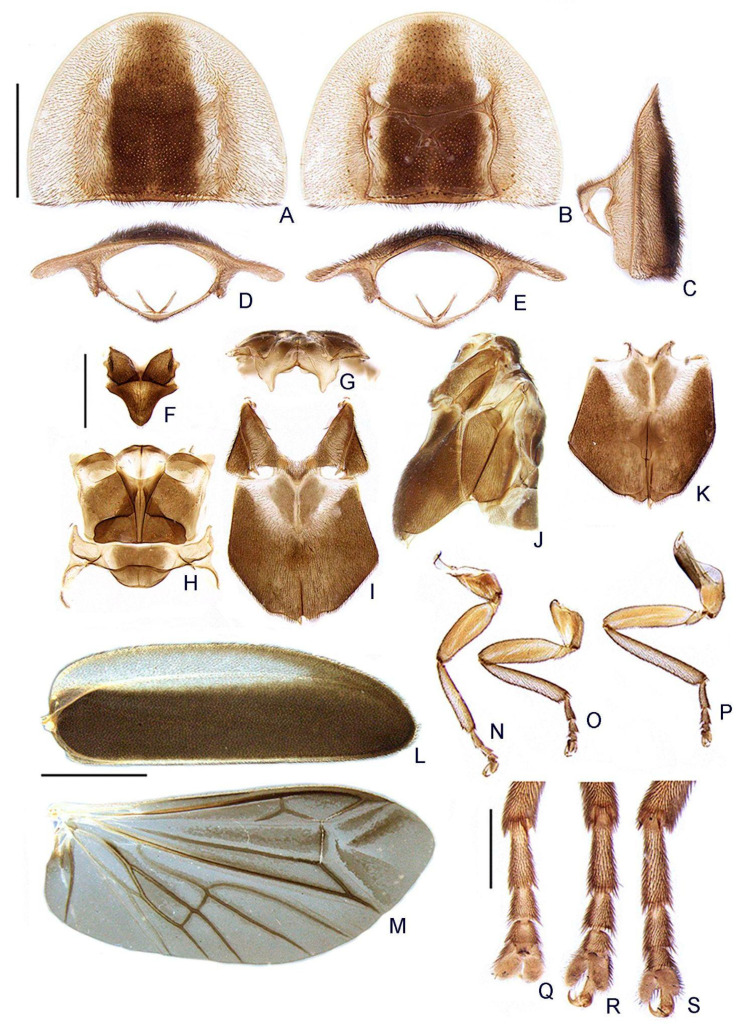
*Haplocauda albertinoi***sp. nov.**, male thorax. Prothorax (**A**–**E**): (**A**) dorsal, (**B**) ventral, (**C**) lateral, (**D**) anterior, (**E**) posterior. (**F**) Mesoscutellum, dorsal. Metanotum (**G**,**H**): (**G**) anterior, (**H**) dorsal. Pterothorax (**I**–**K**): (**I**) ventral, (**J**) lateral, (**K**) detail of meso and metendosternite. (**L**) Elytra, ventral. (**M**) Wing, dorsal. Outline of right legs (**N**–**S**): (**N**) proleg, (**O**) mesoleg, (**P**) metaleg. Detail of the inwards view of right legs (**Q**–**S**)—note the tibial spurs: (**Q**) proleg, (**R**) mesoleg, (**S**) metaleg. Scale bar: A–E = 1 mm; F–K = 1 mm; L–P = 2mm; Q–S = 500 µm.

**Figure 6 insects-13-00058-f006:**
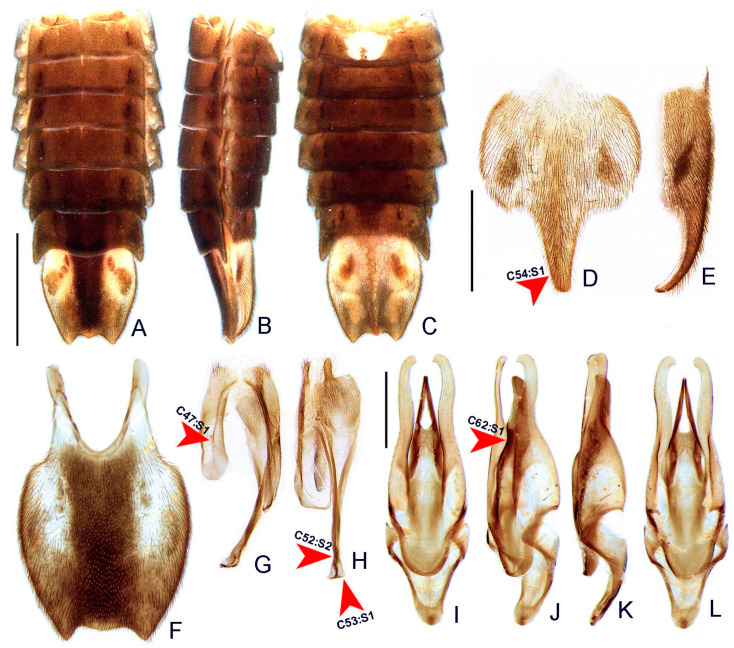
*Haplocauda albertinoi***sp. nov.**, male abdomen. Whole abdomen (**A**–**C**): (**A**) dorsal, (**B**) lateral, (**C**) ventral. Sternum VIII (**D**,**E**): (**D**) ventral, (**E**), lateral. (**F**) Pygidium, dorsal. (**G**) Syntergite, dorsal. (**H**) Sternum IX, ventral. Aedeagus (**I**–**L**): (**I**) dorsal, (**J**) dorso-lateral, (**K**) lateral, (**L**) ventral. Scale bar: A–C = 2 mm; D–H = 1 mm; I–L = 500 µm.

**Figure 7 insects-13-00058-f007:**
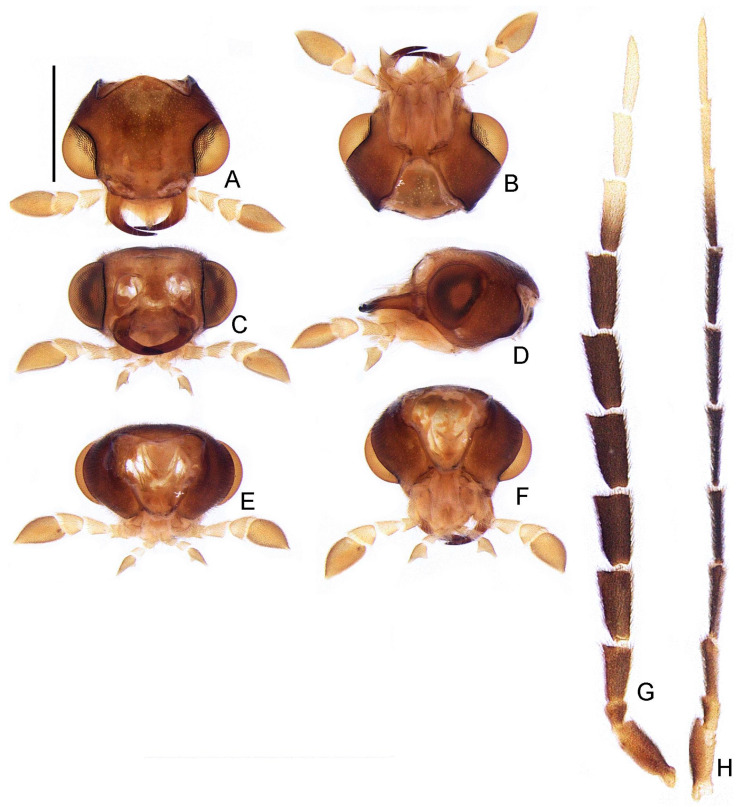
*Haplocauda albertinoi***sp. nov.**, female head. Head capsule (**A**–**F**): (**A**) dorsal, (**B**) ventral, (**C**) frontal, (**D**) lateral, (**E**) posterior, (**F**) occipital. Left antenna (**G**,**H**), (**G**) dorsal, (**H**) frontal. Scale bar: A–H = 1 mm.

**Figure 8 insects-13-00058-f008:**
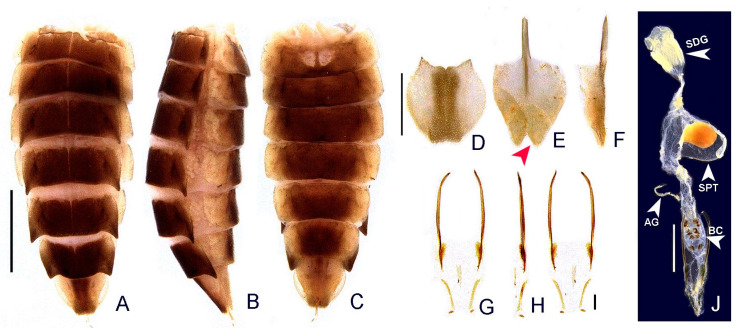
*Haplocauda albertinoi***sp. nov.**, female abdomen. (**A**–**C**) whole abdomen: (**A**) dorsal, (**B**) lateral, (**C**) ventral. (**D**) Pygidium, dorsal. Sternum VIII (**E**,**F**): (**E**) ventral, (red arrowhead indicates moderate indentation), (**F**) lateral. Ovipositor (**G**–**I**): (**G**) dorsal, (**H**) (lateral), (**I**) ventral. (**J**) Internal anatomy of the reproductive tract; SDG: spermatophore digesting gland; SPT: spermatheca; AG: accessory gland. Scale bar: A–C = 2 mm; D–I = 1 mm; J = 1 mm.

**Figure 9 insects-13-00058-f009:**
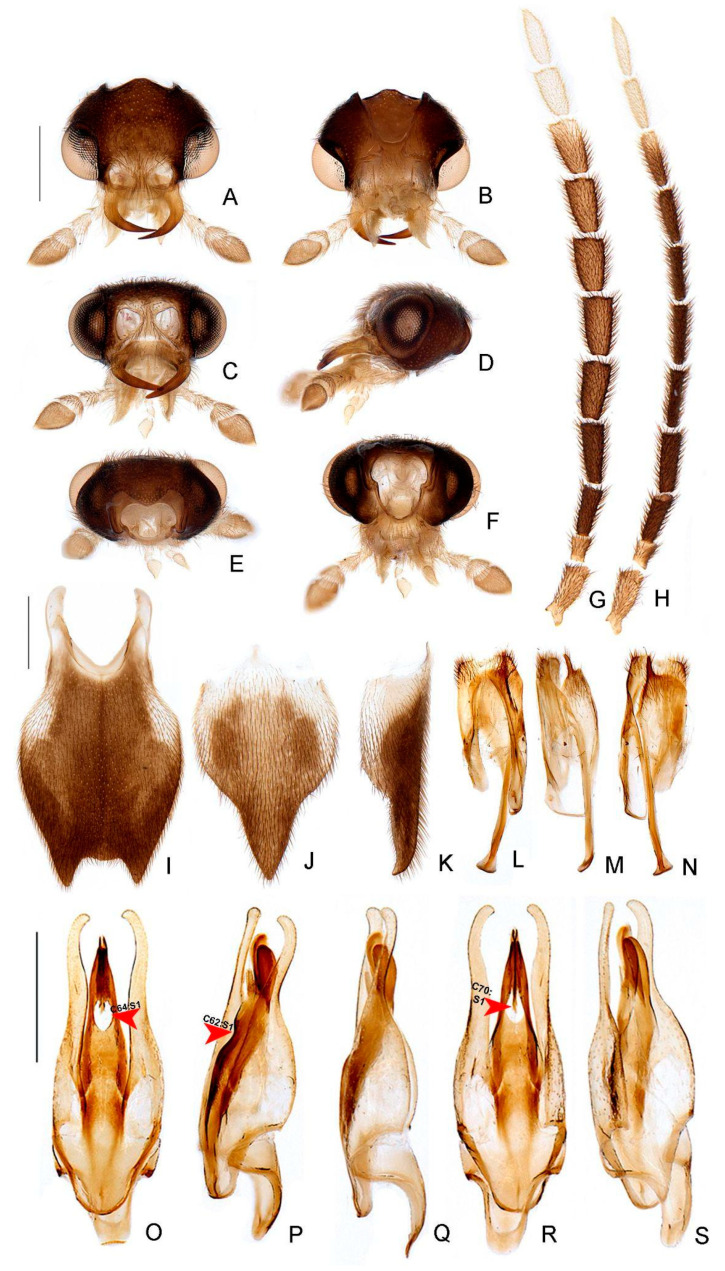
*Haplocauda yasuni***sp. nov.**, male morphology. Head capsule (**A**–**F**): (**A**) dorsal, (**B**) ventral, (**C**) frontal, (**D**) lateral, (**E**) posterior, (**F**) occipital. Left antenna (**G**,**H**), (**G**) dorsal, (**H**) frontal. (**I**) Pygidium, dorsal. Sternum VIII (**J**,**K**): (**J**) ventral, (**K**), lateral. (**L**) Syntergite, dorsal. Sternum IX (**M**,**N**): (M) lateral, (**N**) ventral. Aedeagus (**O**–**S**): (**O**) dorsal, (**P**) dorso-lateral, (**Q**) lateral, (**R**) ventral, (**S**) postero-lateral. Scale bar: A–H = 500 µm; I–N = 500 µm; O–S = 500 µm.

**Figure 10 insects-13-00058-f010:**
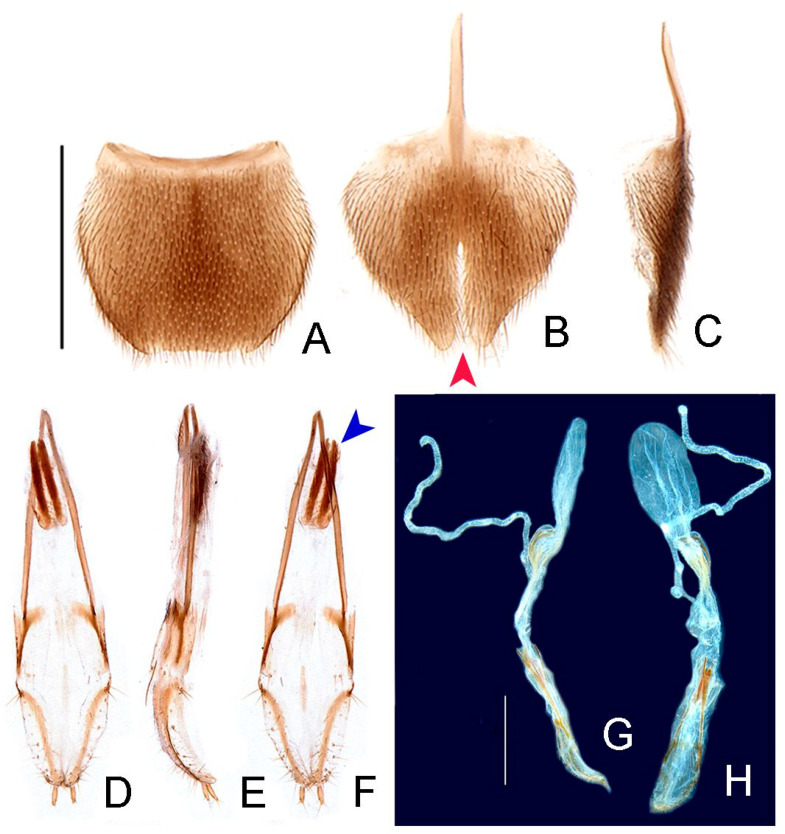
*Haplocauda yasuni***sp. nov.**, female morphology. (**A**) Pygidium, dorsal. Sternum VIII (**B**,**C**): (**B**) ventral, (red arrowhead indicates deep indentation), (**C**) lateral. Ovipositor (**D**–**F**): (**D**) dorsal, (**E**) lateral, (**F**) ventral (blue arrowhead indicates sclerites of the bursa copulatrix). Internal anatomy of the reproductive tract (**G**,**H**): (**G**) lateral, (**H**) ventral. Scale bar: A–E = 1 mm; F–G = 1 mm.

**Figure 11 insects-13-00058-f011:**
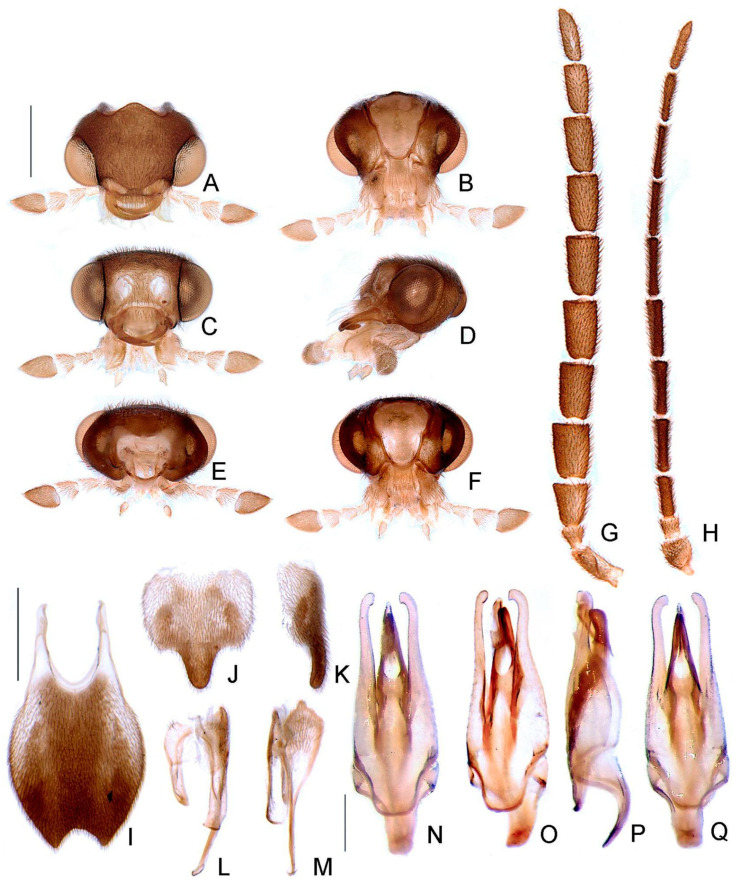
*Haplocauda mendesi***sp. nov.**, male morphology. Head capsule (**A**–**F**): (**A**) dorsal, (**B**) ventral, (**C**) frontal, (**D**) lateral, (**E**) posterior, (**F**) occipital. Left antenna (**G**,**H**), (**G**) dorsal, (**H**) frontal. (**I**) Pygidium, dorsal. Sternum VIII (**J**,**K**): (**J**) ventral, (**K**), lateral. (**L**) Syntergite, dorsal. (**M**) Sternum IX, ventral. Aedeagus (**N**–**Q**): (**N**) dorsal, (**O**) dorso-lateral, (**P**) lateral, (**Q**) ventral. Scale bar: A–H = 500 µm; I–M = 750 µm; N–Q = 250 µm.

**Figure 12 insects-13-00058-f012:**
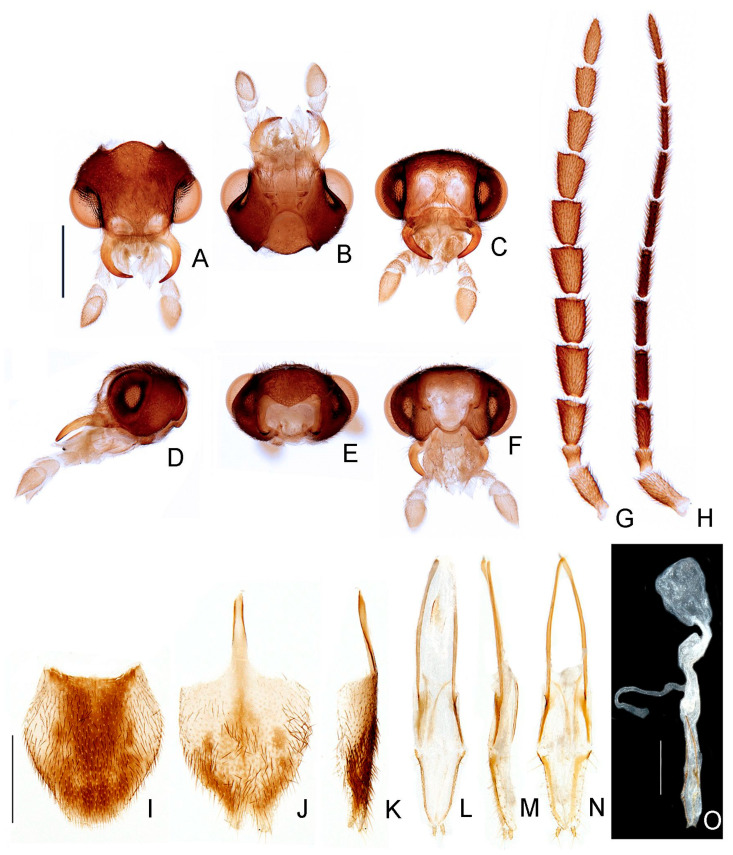
*Haplocauda mendesi***sp. nov.**, female morphology. Head capsule (**A**–**F**): (**A**) dorsal, (**B**) ventral, (**C**) frontal, (**D**) lateral, (**E**) posterior, (**F**) occipital. Left antenna (**G**,**H**), (**G**) dorsal, (**H**) frontal. (**I**) Pygidium, dorsal. Ovipositor (**L**–**N**): (**L**) dorsal, (**M**) lateral), (**N**) ventral. (**O**) Internal anatomy of the reproductive tract. Sternum VIII (**J**,**K**): (**J**) ventral, (**K**), lateral. Scale bar: A–H = 500 µm; I–N = 500 µm; O = 750 µm.

**Figure 13 insects-13-00058-f013:**
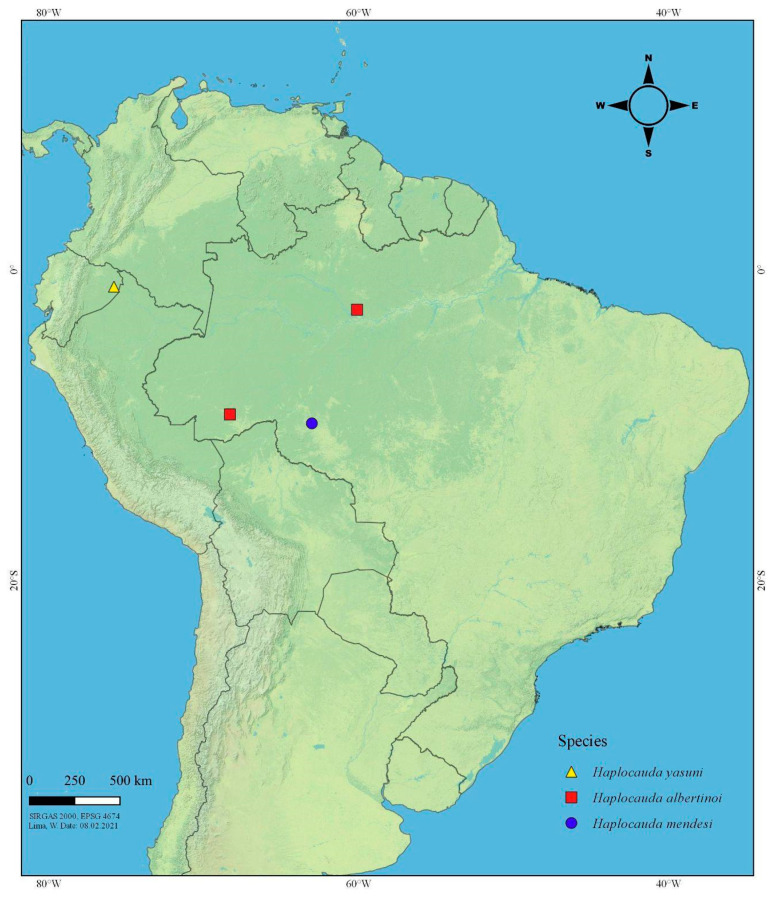
Distribution of *Haplocauda* spp. Note that the species are restricted to the Amazon basin, and are supposedly allopatric.

## Data Availability

Figures in higher resolution, alternative trees and topology test results are available at https://datadryad.org/stash/dataset/doi:10.5061/dryad.w3r2280s7 (accessed on 30 December 2021).

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
