# Peer review of "Haplocauda, a New Genus of Fireflies Endemic to the Amazon Rainforest (Coleoptera: Lampyridae)†"

_insects, 2022, doi:10.3390/insects13010058_

Round 1

Reviewer 1 Report

The manuscript represents an interesting contribution to the systematics of a beetle group. The text is clear, the analyses are correct and the results are fully illustrated. I have just a few comments/ suggestions:

Page 2: line 3: “Australopacific”: meaning?

Page 3: Table I: I don’t think this is a table, this taxonomic info should be incorporated to the text.

Page 30: Diagnosis: I find the generic diagnosis to be extremely long. A few diagnostic characters should be given.

Author Response

Dear reviewer,

Thanks very much for your time and consideration. We will answer your notes below, in bold.

"Page 2: line 3: “Australopacific”: meaning?"

Australopacific is a term used to describe the areas in Oceania plus those in Asia facing the Pacific. It is of common usage, and we don't think it requires explanation in the text, especially since the authors we cite used the word. 

"Page 3: Table I: I don’t think this is a table, this taxonomic info should be incorporated to the text." We took care of it.

"Page 30: Diagnosis: I find the generic diagnosis to be extremely long. A few diagnostic characters should be given." We appreciate your opinion, but we refrain from changing our diagnosis. Given the lack of synthetic, revisionary work in lampyrids in general, having extensive diagnostics is very important.  

Thanks once again,

Luiz et al.

Reviewer 2 Report

The manuscript examine three lampyrid endemic species from Neotropical region, suggesting that they constitute a new monophyletic taxon of generic rank.

The Introduction gives sufficient information about the present status of fireflies of South America. Perhaps authors could also discuss the other species included in the phylogenetic analyses, to furnish a more thorough survey of these taxa.

The M&M section is more than exaustive, since authors also furnished informatin about the recording of the labels. the phylogenetic analyses were correctly chosen and described according to the usual methods applied to morphological datasets.

The results open with the Character List, but no further information is given. The characters were not described in details, but photos are included for some of them. The findings are clearly shown, but the trees can be improved, the quality of the images is low. A single tree is shown. Are the trees resulting form the various analyses identical? If no, how do they differ? This issue could be discussed in the Appendix. The new genus and new species are thorough described. the identification keys have been correctly done.

Discussion and Conclusions are allright.

Please check the references. As far as I know, in the journal the quotes must be included following the orden of quotation, and consequently numbered. Only the numbers must be included in the text. Per quanto ne so, nel diario le citazioni vanno inserite seguendo l'ordine delle citazioni, e numerate. Solo i numeri devono essere inclusi nel testo. Please check the manuscrit for any text and formal errors.

Author Response

Dear reviewer,

We appreciate very much your time and consideration. We will answer your comments below, in bold.

"The Introduction gives sufficient information about the present status of fireflies of South America. Perhaps authors could also discuss the other species included in the phylogenetic analyses, to furnish a more thorough survey of these taxa." Our results are strong in supporting Haplocauda and Scissicauda reciprocal monophyly and close relatedness. However, our nodes were poorly supported beyond that. As such, we didn't discuss the placement of the outgroups in enough detail to warrant a place for them in Introduction, in our humble opinion. If the readers want to know more about these taxa, they can refer to the references.

"A single tree is shown. Are the trees resulting form the various analyses identical? If no, how do they differ? " Actually the tree summarizes information found across ML, BI and parsimony trees. We have added the various trees to the supplemental material. We have discussed the issue in Results, where we had good support to do it.

Thanks once again for your time and consideration.

Best,

Luiz et al.

Reviewer 3 Report

Amazonia has a great diversity of Lampyridae yet highly understudied. The study establish a new genus and three new species of fireflies from Amazonia based on a morphological phylogenetics. The analysis is appropriate, the provided evidence is sounded, and the presentation is good. I would recommend accepting this work for publication for the journal.

Minor modifications are required. They are highlighted in yellow and marked as B (missing in reference) and D (wrong alphabetic order), as well as some capitalization/italic typos. 
Some questions and confusions, which marked as A and C, may need to be answered or improve: 
1. Marked as C: p 7. Several clades have weak support in Fig.1, with low Bremer or bootstrap values. Theoretically the clades are likely clustered by many homoplasies. But in Fig.2 on P9, instead, they are unambiguous synapomorphies (eg. Cladodes + the sister group with 2 characters, (Luciuranus+Uanauna )+sister group with 3 characters). Or vice versa, clades with strong support but are grouped by homoplasies (eg. Araucariocladus+Psilocladus, Vesta+Dilychnia). The authors may need to give more description or explanations on the morphological phylogenetics.  

2. Marked as A: the solid squares in Fig.2 include not only unambiguous synapomorphies but also autapomorphies (e.g. Cladoes flabellatus, Lucidota banoni). The authors may give them different symbols or change figure caption.

3. It is highly suggested providing morphological and anatomic figures of Scissicauda, the sister group of the new genus. The comparison may give readers a good understanding of the diagnostic characters of the new genus.

Author Response

Dear reviewer,

Thanks very much for your time and consideration in reviewing our MS. We will reply to your comments below in bold.

"1. Marked as C: p 7. Several clades have weak support in Fig.1, with low Bremer or bootstrap values. Theoretically the clades are likely clustered by many homoplasies. But in Fig.2 on P9, instead, they are unambiguous synapomorphies (eg. Cladodes + the sister group with 2 characters, (Luciuranus+Uanauna )+sister group with 3 characters). Or vice versa, clades with strong support but are grouped by homoplasies (eg. Araucariocladus+Psilocladus, Vesta+Dilychnia). The authors may need to give more description or explanations on the morphological phylogenetics." The tree shown was the tree with the shortest length. According to the supports found, the addition of a few steps would break down the group sustained by a few unambiguous synapomorphies but not the other one with several homoplasies. This hypothesis is also reflected by the low bootstrap support of the group with a few unambiguous synapomorphies. Since we discuss the supports in the MS, we don't think any further explanation is required.   

"2. Marked as A: the solid squares in Fig.2 include not only unambiguous synapomorphies but also autapomorphies (e.g. Cladoes flabellatus, Lucidota banoni). The authors may give them different symbols or change figure caption." We took care of that.

"3. It is highly suggested providing morphological and anatomic figures of Scissicauda, the sister group of the new genus. The comparison may give readers a good understanding of the diagnostic characters of the new genus". Scissicauda was recently reviewed in an open-access paper, which we refer to in our text. We don't have the rights to the pictures, so we couldn't add these to our paper, but we are confident that by putting both papers together the reader will be able to fully appreciate the differences pointed out in our MS.

As for the low-resolution pictures, that was just for the review process.

Thanks very much for your time and consideration.

Luiz et al.